# University of Nebraska UAS profiling during LAPSE-RATE

Ashraful Islam<sup>1</sup>, Ajay Shankar<sup>1</sup>, Adam Houston<sup>2</sup>, and Carrick Detweiler<sup>1</sup>

<sup>1</sup>NIMBUS Lab, Department of Computer Science & Engineering, University of Nebraska-Lincoln, Lincoln, NE 68588, USA. <sup>2</sup>Department of Earth & Atmospheric Sciences, University of Nebraska-Lincoln, Lincoln, NE 68588, USA.

Correspondence: Ashraful Islam (mislam@huskers.unl.edu)

#### Abstract.

This paper describes the data collected by the University of Nebraska-Lincoln (UNL) as part of the field deployments during the Lower Atmospheric Process Studies at Elevation — a Remotely-piloted Aircraft Team Experiment (LAPSE-RATE) flight campaign in July 2018. UNL deployed two multirotor unmanned aerial systems (UASs) at multiple sites in the San Luis Valley (Colorado, USA) for data collection to support three science missions: convection-initiation, boundary layer transition, and cold air drainage flow. We conducted 172 flights resulting in over 21 hours of cumulative flight time. Our novel design for the sensor housing onboard the UAS was employed in these flights to meet the aspiration and shielding requirements of the temperature and humidity sensors and to separate them from the mixed turbulent airflow from the propellers. Data presented in this paper

include timestamped temperature and humidity data collected from the sensors, along with the three-dimensional position and

10 velocity of the UAS. Data are quality controlled and time-synchronized using a zero-order-hold interpolation without additional post-processing. The full dataset is also made available for download at (https://doi.org/10.5281/zenodo.4306086 (Islam et al., 2020)).

#### 1 Introduction

A team of researchers from the University of Nebraska-Lincoln (UNL) participated in the Lower Atmospheric Process Studies 15 at Elevation — a Remotely piloted Aircraft Team Experiment (LAPSE-RATE) flight campaign between 14 – 19 July 2018 at San Luis Valley of Colorado, USA. LAPSE-RATE was organized as part of the International Society for Atmospheric Research using Remotely piloted Aircraft (ISARRA) 2018 meeting. A total of 1287 flights were conducted by 13 institutions, including UNL, which resulted in more than 260 hours of data collection. UNL's contribution to this collaborative data collection effort was 172 Atmospheric Boundary Layer (ABL) profiling flights using two multirotor UAS platforms. These flights from UNL

- resulted in over 21 hours of data being collected. This unique collaboration resulted in a collective sampling of a variety of atmospheric phenomena over the span of six days at preplanned sites around the San Luis Valley. An overview of the LAPSE-RATE campaign, the description of site locations, and science missions that focused on measuring different atmospheric phenomena of interest are documented (de Boer et al., 2020a, b). Data from UNL and all other participating teams in the LAPSE-RATE campaign are hosted in an open access data repository (LAPSE-RATE Data Repository, 2021).
- Multirotor UASs are finding more routine uses for sampling and profiling the ABL, such as atmospheric profiling (Bonin et al., 2013; Elston et al., 2015; Greatwood et al., 2017; Jacob et al., 2018; Islam et al., 2019; Barbieri et al., 2019; Segales

et al., 2020), estimation of the spatial structure of temperature (Hemingway et al., 2020), wind measurement (Prudden et al., 2016; Palomaki et al., 2017), and prediction of Lagrangian coherent structure (Nolan et al., 2018).

The need for increased spatial resolution for atmospheric sampling is reflected in publications, such as improving Numerical Weather Prediction (NWP) models (Leuenberger et al., 2020), improvement of mesoscale atmospheric forecast (Dabberdt et al., 2005), and identification of hazardous weather for Beyond Visual Line of Sight(BVLOS) flights using UAS Traffic Management (UTM) systems (Mitchell et al., 2020). UASs can meet such profiling needs with a greater frequency of profiles, increased spatiotemporal resolution of data, and sampling in virtually any sampling location when compared with traditional methods. Multirotors extend the sampling capability by allowing rapid and repeatable profiling at any site while maintaining a

35 fixed horizontal position.

Our previous work (Islam et al., 2019) describes the design and evaluation of a temperature and humidity (TH) sensor housing that meets the recommended sensor placement, aspiration, and shielding criteria by using a passively induced-airflow technique that works by exploiting the existing UAS propeller. The housing's inlet is pointed outwards from the UAS to sample just outside of the UAS turbulence in both ascent and descent. This is different from existing methods of placing the

- sensor under the arm without shielding but aspirated by the propeller (Hemingway et al., 2017), on the body of the UAS without shielding and aspiration (Lee et al., 2018), on a different part of the UAS with shielding and possible aspiration from propellers (Greene et al., 2018) or shielding the sensor inside UAS and active aspiration using a fan while pointing the inlet towards the wind (Greene et al., 2019). All of these existing configurations fail to produce reliable data during descent, and these data are usually discarded (Lee et al., 2018). As multirotor flight time is very limited, needing to discard entire descent
- data prevents optimal use of resources. Additionally, in most cases, observations are affected by wind direction and require onboard sensing of wind and reorientation of UAS with the change of wind direction (Greene et al., 2019).

Two primary highlights of our novel sensor housing are its ability to obtain temperature and humidity sensor readings reliably during both ascent and descent profiles, and its invariance to the aircraft orientation relative to the ambient wind. Two key design considerations in achieving these goals are: the placement of the sensor, and its consistent aspiration. Placement

- of the sensor on the UAS body can adversely affect the measurements (Greene et al., 2018; Jacob et al., 2018). As observed through prior experimental results (Villa et al., 2016), the accuracy of a sensor's measurement increases the farther away it is placed from the propeller's downwash. More specifically, a sensor placed at a distance at least 2.5 times the propeller diameter away from the rotor experiences significantly less aerodynamic interference (Prudden et al., 2016). Consistent and sufficient aspiration is also necessary for a consistent effective sensor response time (Houston and Keeler, 2018). Placing the sensor
- inside the propeller region or near the body can result in inconsistent aspiration due to rotor turbulence (Diaz and Yoon, 2018; Yoon et al., 2017). As such, we designed our sensor housing to source the sampling air from outside rotor interference and to maintain consistently high aspiration airspeed to obtain reliable results.

Our sensor housing design has evolved over multiple design iterations and has been field-tested in multiple Collaboration Leading Operational UAS Development for Meteorology and Atmospheric Physics (CLOUD-MAP) field campaigns (Jacob

et al., 2018). The details of our data validations tests, as well as a complete description of the sensor housing design, are available in a separate open access paper (Islam et al., 2019).

**Figure 1.** Images of (A) the UAS setup with the temperature and humidity sensor mounted in the aspirated and shielded sensor housing, and in a traditional configuration (B) Close up of the traditionally mounted sensor under the UAS without the sensor housing (inside the white circle), (C) Close up of the sensor housing with the sensor mounted, and (D) Close up of the sensor probe mounted inside our sensor housing. The outlet of the sensor housing is placed on top of the propeller, and the inlet is pointing outward. High-speed air in the sensor housing is drawn passively by exploiting the pressure deficit created by the propeller of the UAS.

For the LAPSE-RATE campaign, UNL deployed two identical UASs with one primary sensor suite for measurements, and a secondary sensor suite for redundancy and testing. These flights were conducted at five locations in San Luis Valley (Colorado, USA) through 14–19 July 2018. The maximum altitude for each flight ranged from 100 – 500 m above ground level. Figure 1
illustrates the UAS with the housing setup, closeup of the sensor housing, and sensor mounting configurations. Both primary and secondary sensors are located inside their respective sensor housings mounted on two diametrically opposing arms of the UAS. In some flights, a third sensor was mounted *under* the body frame of the UAS to compare the performance of primary sensors against traditional mounting positions. A detailed description of our configuration is presented in Section 2.5. It should be noted that, although the data collection is focused on the temperature and humidity measurements, atmospheric pressure

data from the sensors are also included in the dataset for anyone interested.

The rest of the paper describes the components of our system (Section 2), the flight strategies employed for missions (Section 3), the data processing used (Section 4), and some special topics of interest (Section 5). We finally conclude with an example profile data, and provide details regarding the availability of the dataset.

#### 2 System Description

#### 75 2.1 UAS platform

The two identical UASs deployed during the missions were developed on a DJI Matrice 600Pro platform equipped with DJI A3 Pro flight control systems. The unfolded dimensions (including propellers, frame arms, GPS mounts, and landing gear) of the system are  $1668 \text{ mm} \times 1518 \text{ mm} \times 727 \text{ mm}$ . The recommended maximum payload capacity of the platform is 5.5 kg. At no load, the UAS has a flight endurance of 35 - 40 min on a single set of six DJI TB48S batteries. The manufacturer-specified positioning accuracy is  $\pm 0.5 \text{ m}$  in the vertical axis, and  $\pm 1.5 \text{ m}$  horizontal (DJI, 2021b). The maximum ascent and descent speeds are  $5 \text{ m s}^{-1}$  and  $3 \text{ m s}^{-1}$ , respectively. The flight controller offers real-time access (read-only) to UAS's onboard sensor data, such as position, velocity, and attitude, through a serial interface. Additionally, a mobile application allows a user to plan and deploy a flight trajectory, and the remote controller allows intervention from the user at any point.

#### 2.2 Sensors


- Table 1 describes the specifications of the temperature and humidity (TH) sensors used for the dataset. Every UAS flight used one iMet XQ2 from InterMet Systems (Grand Rapids, MI, USA) as the primary TH sensor. The XQ2 is a self-contained sensor package designed for UASs to measure atmospheric pressure, temperature, and relative humidity. It is also equipped with a built-in GPS and an internal data logger along with a rechargeable battery. A serial interface provides access to the logs, or real-time observations produced by the sensor at 1 Hz. The internal data-logger was only used as backup and is not part of this
- dataset. Data included in the dataset were collected through the data acquisition (DAQ) system using the serial interface. Some UAS flights featured an older version of this sensor, called iMet XQ1, as the secondary backup sensor.

Some flights also used a nimbus-pth as the secondary sensor, which is a sensor package unit we designed and built for pressure, temperature, and humidity sensors. Several nimbus-pth can be chained as nodes for data collection. In some data files, two of these nodes might be present. In such cases, one of them was aspirated inside our sensor housing, and the other

one sat directly underneath the UAS in a traditional non-aspirated configuration. In the data files, the first two sensors were shielded and aspirated inside the housing, and the third sensor (when available) was in a traditional non-aspirated configuration.

#### 2.3 Sensor Housing

The sensor housing is designed to meet or exceed sensor placement requirements, such as consistent aspiration for the sensors, shielding from solar radiation and other indirect heat sources. The housing draws air passively by exploiting the pressure differential between the region just above a propeller and the region just beyond the rotor wash. The airflow through the

**Table 1.** The key manufacturer's specifications for the sensors used in different experiments: The unavailable fields are left blank. Datasheet for each sensor packages are available at iMet XQ2 (InterMet Systems, 2021b), iMet XQ1 (InterMet Systems, 2021a), and nimbus-pth (Digikey, 2021; Mouser, 2021)

|             |               | XQ2                                                          | XQ1                                  | nimbus-pth                        |  |
|-------------|---------------|--------------------------------------------------------------|--------------------------------------|-----------------------------------|--|
|             |               | (iMet XQ2)                                                   | (iMet XQ1)                           | (Custom Built)                    |  |
| Temperature | Туре          | Bead Thermistor                                              | Bead Thermistor                      | Bead Thermistor                   |  |
|             | Range         | $-90\mathrm{to}50^{\circ}\mathrm{C}$                         | $-95\mathrm{to}50^{\circ}\mathrm{C}$ | $-40$ to 100 $^{\circ}\mathrm{C}$ |  |
|             | Response Time | $1 \mathrm{s} @ 5 \mathrm{m} \mathrm{s}^{-1}$ $2 \mathrm{s}$ |                                      |                                   |  |
|             | Resolution    | $0.01^{\circ}\mathrm{C}$                                     | $0.01^\circ\mathrm{C}$               | $0.01^{\circ}\mathrm{C}$          |  |
|             | Accuracy      | $\pm0.3^{\circ}\mathrm{C}$                                   | $\pm0.3^{\circ}\mathrm{C}$           |                                   |  |
| Humidity    | Туре          | Capacitive                                                   | Capacitive                           | Capacitive                        |  |
|             | Range         | $0-100\%\mathrm{RH}$                                         | $0-100\%\mathrm{RH}$                 | $0-100\%\mathrm{RH}$              |  |
|             |               | @ 25 °C, 0.6 s $5 \text{ s}$ @ $1 \text{ m s}^{-1}$ veloc    |                                      | 8 s                               |  |
|             | Response Time | @ 5 °C, 5.2 s                                                |                                      |                                   |  |
|             |               | @ $-10$ °C, 10.9 s                                           |                                      |                                   |  |
|             | Resolution    | $0.1\%\mathrm{RH}$                                           | $0.7\%\mathrm{RH}$                   | $0.01\%\mathrm{RH}$               |  |
|             | Accuracy      | $\pm5\%\mathrm{RH}$                                          | $\pm5\%\mathrm{RH}$                  | $\pm2\%\mathrm{RH}$               |  |

housing is always maintained as long as the propellers are spinning, and provides a consistent aspiration for the sensors (Islam et al., 2019). The inlet and outlet of the housing are shaped like a cone to provide high-speed airflow across the housing tube with a small pressure difference between the two ends. Additional design considerations are made to ensure that the flow is consistent, and provides airflow  $\geq 5 \text{ m s}^{-1}$  across the sensors even at the lowest propeller speeds.

Sensors are placed inside the tube structure as shown in the panel C and D of Figure 1. The entire sensor housing is painted with reflective white paint, and tubes are wrapped with aluminum foil tape. This results in excellent rejection of solar heating and avoids unpredictable radiation heating bias. Such placement of sensors provides solar shielding and shielding from other artificial heat sources such as motor or battery waste heat. Since the entire housing is placed outside the body of UAS, it creates further isolation from the artificial heat sources in the UAS. Additionally, since the aspirating airspeed is very high (Islam et al., 2019), it reduces the error from all these sources even further (Anderson and Baumgartner, 1998).

The housing is also designed to be modular, printed entirely using a 3D printer, and has an easy screw-in assembly. The impact of the housing on the UAS's stability and flight time is minimal. Further details and the full schematic of the housing and the evaluation can be found in our previous work (Islam et al., 2019).

#### 2.4 Data acquisition

Data were collected using a data acquisition (DAQ) system comprised of an Odroid XU4 (Hardkernel, 2021), a compact single-board computer that runs a Linux operating system. Odroid runs the robot operating system (ROS) (Quigley et al.,

2009) that communicates with the serial devices through its USB ports. ROS facilitates collecting many different sensor data independently at their own output frequency, recording the timestamp for when data were generated and when they are received by ROS. ROS interfaces the collection of all available devices even in the case of a single device failure. Synchronization of the data can either be done at runtime or in post-processing. In our case, it was done in post-processing using MATLAB.


The communication with the DJI flight controller was implemented using the ROS interface of DJI Onboard SDK (DJI, 2021c) available to developers. This allowed the recording of all the telemetry data from the flight controller, along with high-quality positioning information. The GPS data from the iMet XQ2/ iMet XQ1 sensor were discarded as the positioning information from the flight controller was found to be of better quality.


The Odroid was connected with a ground computer using wireless 2.4 GHz XBee radios for the operation of DAQ, debugging, and periodic checks on the data when the UAS finished a flight. The data collected by the DAQ were retrieved to the ground computer for archiving at the end of each day using an ethernet connection.

Temperature and humidity sensors were connected over serial with ROS to send periodic updates of the observations. The UAS's autopilot also interfaced with ROS to provide updates of position, velocity, altitude, attitude, etc. which were also recorded to spatially and temporally synchronize the observation.

## 2.5 UAS Sensor Mounting Configuration and Payload

As mentioned in Subsection 2.2, the primary sensor was the iMet XQ2, and its data were recorded on the dataset with a header underscore \_1 (e.g., Temperature\_1, Humidity\_1, Pressure\_1). Other sensor data headers were followed with \_2 and \_3 when available. Sensor\_1 and Sensor\_2 were shielded inside the sensor housing; however, sensor\_3 was placed under the UAS in a traditional configuration without aspiration. The placement of the sensors inside the housing and sensors without the housing are marked in Figure 1 for reference. Specific placements of the sensors on the UAS used in the data collection are described below.

#### 2.5.1 UAS platform M600P1

One XQ2 (sensor\_1) was placed inside the left sensor housing, and one XQ1 (sensor\_2) was on an identical right sensor
housing. This placement location for the left housing is highlighted in the 'panel (A)' of Figure 1. The alternative setup used in some experiments replaced XQ1 with nimbus-pth (sensor\_2) inside the right sensor housing (sensor names are also listed in metadata as data source). If nimbus-pth is included in measurements, it was placed under the body of the UAS without any housing structure, as highlighted in the 'panel (A)' and 'panel (B)' of Figure 1.

#### 2.5.2 UAS platform M600P2

One XQ2 (sensor\_1) was mounted inside the left sensor housing, one nimbus-pth (sensor\_2) was mounted inside the right sensor housing, and an additional nimbus-pth (sensor\_3) was placed under the body of the UAS without a housing. This form of sensor placement facilitates an evaluation between the sensor placed inside the housing versus under the body of the UAS

**Table 2.** Latitude, longitude, and mean sea level (MSL) altitude of operation locations in World Geodetic System 84 (WGS 84) decimal degrees.

| Location       | Latitude  | Longitude   | Altitude (MSL)   |  |
|----------------|-----------|-------------|------------------|--|
| Golf           | 37.626963 | -105.820028 | $2298\mathrm{m}$ |  |
| Gamma          | 37.893536 | -105.716137 | $2329\mathrm{m}$ |  |
| Leach Airfield | 37.784560 | -106.044552 | $2316\mathrm{m}$ |  |
| India          | 38.051294 | -106.102885 | $2332\mathrm{m}$ |  |
| Charlie        | 38.052690 | -106.087414 | $2329\mathrm{m}$ |  |

without housing. It also allows comparison of the sensors mounted on the opposite ends of the UAS. Having secondary sensors also provides a fail-safe when the primary sensors fail, such as the case on XQ2 humidity sensors on 17 July 2018 data.


The UAS's total payload during the experiments was approximately 1.8 kg. Two sensor housings with their support structure and sensor were approximately 720 g each; the onboard computer was 140 g; and misc cables, screws, etc., were approximately 200 g. UAS flight endurance was 20 - 25 min with the payload.

#### **3** Flight locations and strategies

#### 3.1 Flight locations

- During the LAPSE-RATE field campaign, measurement objectives for each day were determined based on the weather forecast, site availability, and available team resources. Many designated locations of San Luis Valley of Colorado, USA, were planned beforehand as atmospheric sampling sites depending on atmospheric phenomena of interest. The planning of locations, atmospheric phenomenon to be observed for the day, and assignment of teams are described in (de Boer et al., 2020a). We conducted flights in locations designated as Golf, Gamma, Leach, India, and Charlie between 14 – 19 July as part of the LAPSE-RATE flight campaign (de Boer et al., 2020b) as well as individual research objectives. GPS coordinates of these loca-
- tions are provided in Table 2 and illustrated in a terrain map in Figure 2. The 'inset (B)' of Figure 2 shows the flight locations of UNL UASs in the context of all the LAPSE-RATE flight campaign locations of interest where all the teams were operating based on the measurement objective of the day.

#### **3.2** Flight strategies

Flight strategies for each day were dictated by atmospheric phenomena being measured. The teams participating in the LAPSE-RATE campaign coordinated flights across the San Luis Valley according to the atmospheric phenomena of interest for the day and the atmospheric variability expected at different sampling locations. Measurement objectives of LAPSE-RATE in which UNL participated in data collection are calibration flight (CLF), boundary layer transition (BLT), convection initiation (CI),