# Peer review of "University of Nebraska UAS profiling during LAPSE-RATE"

_Earth System Science Data, 2020_

## Referee Comment (RC1) · Anonymous Referee #1 · 11 Jan 2021

General Comments: This data paper highlights contributions from researchers at the University of Nebraska-Lincoln to the 2018 LAPSE-RATE Campaign. The paper is well organized and provides sufficient explanation of the hardware used in data collection. The sensor deployment used to collect the data is relatively unique among LAPSE-RATE participants.

Specific Comments: Line 17 - Define what fixed-site profiling means here. I believe the intent is that multirotors can fly to and remain at a fixed point for a period of time. Line 20 - Define CLOUD-MAP before first use. Line 50 - Why weren't the barometric pressure sensors integrated in the iMet-XQ2 and nimbus-pth sensors included in the technical description? Line 82 - Elaborate on what periodic checks of the data means. I assume you mean a human is observing that data are being collected and the values

appear to be reasonable. Were there instances where you observed abnormal data collection mid-flight using the wireless data stream and modified or aborted a flight, or switched out instrumentation after a flight? Line 83 & 84 - Was the interface between the DJI M600P flight controller and the Odroid a turn-key solution or did you have to develop any custom software to decode the DJI telemetry stream? Were GPS data from the iMet-XQ2/1 discarded? Line 111 - Define MURC before first use.

Technical Corrections: Line 83 - add "The" in front of "UAS's". Lines 102 & 103 - Spell out approximate instead of approx. Replace $\sim$ with the same term for consistency. Use "g" instead of "gm" for abbreviated units of grams.

Check the journal spacing requirements when using units. Spacing is inconsistent throughout the manuscript and tables.

Some minor editing for consistent uses of past tense is needed.

---

## Referee Comment (RC2) · Anonymous Referee #2 · 18 Jan 2021

**Recommendation:** Accept with major revisions

**General Comments**

This data overview paper outlines and describes the rotary-wing UAS data collected by the University of Nebraska-Lincoln in the 2018 LAPSE-RATE campaign. The writing is clear and concise and the paper is decently well structured. There are a few aspects that I would like to see improved before publication, and I move to accept with major revisions to enhance the details of this paper.

I think that the discussion of the hardware (Section 2), though concise, was well handled and provides a thoughtful overview of the system utilized for data collection. However, more information about the logistics of data collection (Section 3) would be nice.

The biggest complaint that I have with this paper is that it is lacking in context and specifics. This paper feels detached from the special issue's context. It currently does not even reference the campaign's overview article in this special issue or the Bulletin of the American Meteorological Society that would help provide the missing backdrop for this data set. Please make a greater effort to tie in your work to the context of the larger effort.

In general, there are a few stylistic points that could be improved as well. Most figure captions are also lacking in detail that could help to better inform the reader about the purpose of including the figures. This issue should also be addressed in the main text by discussing the figures and their significance more. Moreover, please be sure to follow the ESSD journal conventions for including numbers and units (see here: https://www.earth-system-science-data.net/submission.html#math). Please see the points below for more specific instances of these recommendations.

**Major Comments**
- Section 1, Introduction: in general, this paper is missing the context of being part of the larger LAPSE-RATE campaign, which should be improved by including references and discussion for at least the following:
    - de Boer et al. (2020a): https://doi.org/10.1175/BAMS-D-19-0050.1
    - de Boer et al. (2020b): https://doi.org/10.5194/essd-12-3357-2020
- Sections 1 and 2.3: while the authors do a good job of discussing their sensor housing setup, it is also important to include references to other studies that have performed similar work to provide context to someone trying to use this data that

may or may not be familiar with UAS sensor housing and limitations. I would therefore like to see the following:

- ○ Provide more discussion/details here from the the Islam et al. (2019) paper to better contextualize this specific aircraft.
- ○ Depending on the specific details you include from the Islam et al. (2019) study, it would also be beneficial to elaborate more on the Villa et al. (2016), and Prudden et al. (2016) studies.
- ○ More context could be added by including and possibly briefly discussing the Greene et al. (2019) study (https://doi.org/10.3390/s19061470), which is effectively a continuation of their 2018 study you already cited and is more closely related to the applications of the sUAS discussed in this ESSD paper.

- ● Section 3.1 (L 107): The reader has no context for what these sites are without the introduction of the LAPSE-RATE campaign as a whole, which is currently missing from Section 1. Please provide proper context (and citations) to how these sites fit with the larger campaign as well as some details about them.
- ● Section 3.2: I think this section would benefit from further organization, more specifically by splitting each day into subsections and provide more details per day (please look at the other accepted/published papers in this ESSD special edition).
- ● L 142: You mention using a "zero-order-hold-method" here as your main data processing method. Please elaborate what this method is and how you applied it here in constructing your data files.
- ● Section 3: In Line 124, you mention you could not go to altitudes above 120 m because the NOTAM was not active. This sentence is a bit misleading because there is no mention of how else you accessed the airspace and generally the NOTAM is just the proof the request was filed properly. This section would benefit from what permissions you relied on (COA or 107 exemption) and what your maximum allowed altitude was. With more people getting into sUAS work, it's important for people within our community to be transparent about how to legally do this work.
- ● Section 5 Special topics of interest (L 152): I very much enjoy the presentation of how this data set can be used to examine broader questions of platform and sensor performance. However there is little to no context as to why these topics are important for people outside our community who might be interested in using this as sounding data. I suggest you provide few sentences of background as to why each of these are important before saying which flights may be utilized to examine this phenomena. For example, what are some of the challenges

associated with not optimizing ascent/descent speeds? Wind direction versus sampling?

- Figures 4 and 5: I'm not sure I understand the utility of presenting the data in this manner. Especially in the humidity sensors you seem to be having a fair bit of hysteresis. Please provide discussion as to whether this is an accurate depiction of the environmental variability or if it more closely linked to sensor hysteresis.

**Minor Comments**

- L 8: replace "temperature/humidity sensors" with "temperature and humidity sensors"
- L 8: I believe there is a subject/verb disagreement in "...attempt to separate them…"
- L 14-15: more references providing examples of how multirotor UASs are gaining popularity would be useful
- L 16-16: What are some examples of applications that would benefit from sounding data with increased spatiotemporal resolution? Additional references and specifics here would be nice.
- L 20: please define the acronym "CLOUD-MAP"
- L 25: what do you mean by "validity of the measurement"? Please elaborate.
- L 39: to conclude the introduction, please include a general outline for how the rest of the paper is organized.
- L 44: there should be a space between the number and units; "mm" should not be italicized
- L 47: change instances of "m/s" to use a "-1" exponent instead
- L 50, Section 2.2 header: remove the colon at the end
- L 54: remove the comma after "GPS" at the start of the line
- L 55: insert a space between "1 Hz"
- L 56: please define the acronym "DAQ" as this is the first instance of use
- L 61: insert "the" so it reads: "In the data files, the first two sensors…"
- L 70: unit conventions on 5m/s
- L 79: 'Fail' should be 'failure'.
- L 81: insert a comma to read as: "...DAQ, debugging, and periodic…"
- L 84: 'Interface' should be 'interfaces'.
- L 90: please connect these sensor descriptions back to the diagram of the UAS in Figure 1 for reference.
- Section 2.5 in general: please make the boldface headers into subsubsections. For example, L 91 should change to: "2.5.1 UAS platform M600P1", etc.
- L 100: add an "s" onto the end of "sensor" to read as: "It also allows comparison of the sensors mounted on…"

- L 101: Add a comma after "when the primary sensors fail,"
- L 101: Remove the comma in the date at the end of the line to read as "17 July 2018".
- L 103: fix the unit conventions for "grams"
- L 105, Section 3 header: remove the comma after "locations"
- L107: add an "and" and "the" to read as: "...Leach, India, and Charlie in the LAPSE-RATE flight campaign".
- L 113-114: "ascended to the height of the MURC tower." How tall is that?
- L 113-114: fix the unit conventions for m, m/s, and seconds.
- L 119: fix the unit conventions for m and m/s
- L 119: change "in Golf and Gamma location" to "at the Golf and Gamma locations"
- L 120-121: Please split up the weather descriptions here and for the other days into multiple complete sentences.
- L 124: fix unit conventions for m and m/s
- L 127: add an "s" at the end of "condition"
- L 141: add "the" in front of "UAS flight controller"
- L 143: "1 second" should be "1 s" to be consistent with unit conventions
- L 150: please describe the file conventions in detail here even though it is also included in the README file to be consistent with the other published/accepted papers in this ESSD special issue.
- Section 5 in general: please specify subsections for the bold headings and remove the colons after the headings. For example, L 154 should read as: "5.1: Calibration".
- L 160: date format earlier in the paper was DD month YYYY; please change to be consistent here
- L 161: add a local conversion to MDT from UTC
- L 161-162: unit conventions for m and m/s
- L 165: unit conventions for m/s
- L 171-173: this information more appropriately belongs in the caption for Figure 3
- L 174: change wording at the beginning to be: "Figures 4 and 5 show primary sensor…"
- L 195, References section: please alphabetize your references
- Table 3: No. of Flight should be plural
- Table 3: Please make more noticeable if multiple aircraft are at the same location for a day (e.g., add an "&" in between them)
- Figure 1: additional close-up photos of the shields and under-body should be included to give the reader a better spatial understanding of the UAS sensor payload

- Figures 3-5: It would be helpful to label your panels A, B, C, D, etc. and then further describe the nuances in the captions.

---

## Author Comment (AC1) · 30 Apr 2021

**University of Nebraska UAS profiling during LAPSE-RATE**

Ashraful Islam, Adam Houston, Ajay Shankar, Carrick Detweiler

**Outline of the responses to Referee comments:**

- 1. Response to Topical Editor comments (Page 1-4)
- 2. Response to Anonymous Referee #1 comments (page 5-7)
- 3. Response to Anonymous Referee #2 comments (page 8-17)

**Response to Topical Editor Comments:**

**Topical Editor summary of the paper:**

The paper summarizes the thermodynamic data sets from two rotorcraft platforms operated by the University of Nebraska-Lincoln (UNL) in support of the 2018 LAPSE-RATE campaign. Thermodynamic sensors are mounted inside a custom passive aspiration and protective housing, and in some cases include a third externally mounted sensor for comparison. Locations, times and plotted temperature and humidity measurements for each day's observations are presented, along with discussion of data processing.

**Authors response:**

Thank you for your detailed feedback and comments to improve the quality of the paper. We have made the suggested changes and corrections to the paper.

**Citation issues:**

1. pg 5, line 107: add reference to special issue LAPSE-RATE overview de Boer, et. al.

de Boer, G. et. al, (2020) "Data Generated During the 2018 LAPSE-RATE Campaign: An Introduction and Overview," Earth Syst. Sci. Data, https://doi.org/10.5194/essd-2020-98.

- 2. pg 5, line 111: You included a reference to the MURC website, but didn't cite it in the text that I found. This would be a good place for it.
- 3. pg 6, line 117: Not sure what you mean by your citation, "LAPSE-RATE Community, 2020." If this is the BAMS paper, it's not included in the reference list.

de Boer, G., et. al. (2020) "Development of Community, Capabilities and Understanding through Unmanned Aircraft-based Atmospheric Research: The LAPSE-RATE Campaign," BAMS-D-19-0050, *Bull. Amer. Meteor. Soc.*, **101** (5): E684–E699. https://doi.org/10.1175/BAMS-D-19-0050.1

- 4. pg 12-13, References: revise in alpha order.
- 5. pg 12, line 204: not cited (Houston, 2012)
- 6. pg 12, line 214: not cited (MURC)
- 7. pg 12, line 222: not cited (mobile surface vehicles)

Authors response:

Thank you for pointing out the above citation issues. Missing citations are added in relevant texts, and the references are now revised in alpha order.

**Grammar and clarity of text:**

- 1. pg 7, Table 3 Headings: No. of Flights (add s in Flights)
- 2. pg 8, line 164: the first six flights (add s in flights)
- 3. pg 9, line 175: Add that it's "plotted using an artificial horizontal axis offset for clarity." Also correct "figures serve" (remove the s in serves).
- 4. pg 10, line 181: Remove redundant sentence, "All data are available ..."

Authors response:

Thank you for pointing out the grammatical errors and suggestions for improving clarity. We have corrected the errors and improved the clarity in suggested lines of text.

pg 9, Figure 3 and line 173: Comment on the discrepancy between ascent and descent RH at low altitude in the plots presented in Figure 3. If the housing is designed to address ascent/descent differences, why are these different?

Authors response:

We have clarified the source of discrepancy between ascent and descent RH and Temperature in Figure 3(now Figure 4). We have expanded the captions with more information to aid the synthesis of information. We have added the following texts after line 173 (now line 325):

"Although the housing is designed to address ascent/descent differences, the sensor and the housing have an inherent response time that can not be eliminated. The utility of the presented sensor housing is to keep the effective response time consistent irrespective of the atmospheric condition or orientation of the sensor relative to the wind/sun. The data presented in the figures are not filtered or corrected for effective sensor response time. The raw data without any sensor response correction is presented to show the impact of proper sensor housing on the observations collected by temperature and humidity sensors. This response lag causes a deviation in ascent/descent reading as is expected. Ascent/descent deviation for humidity sensor is larger due to its slower response time in colder temperatures. Even without any correction, ascent and descent readings in our data were within the bounds of sensors uncertainty ( $\pm 0.3 \circ C$  and  $\pm 5 \%$  RH) for temperature and humidity sensors, respectively) and show how effective sensor housing is in collecting quality data. It should be noted that correction can be done using sensor response time as listed by the manufacturer in Table 1. A rigorous correction would require the characterization of the sensor installed in the housing 'as flown' (McCarthy, 1973). The data from MURC (de Boer et

al., 2020c) and UNL Mobile Mesonet (de Boer et al., 2020c) can be used as an additional calibration point, as discussed in Section 5."

pg 9, Figures 4&5 and line 176: Comment on results to notice in Figures 4 and 5. Also could comment on the discrepancy between ascent and descent RH at low altitude in the plots presented in Figure 5 (there are a number of ascent/descent differences).

**Authors response:**

We have added results to notice for figure 4 and 5 (now 5 and 6). We have expanded the captions with more information to aid the synthesis of information. We have also added discussion of potential sources of discrepancy between ascent and descent RH at lower altitudes. Additional texts that were added:

"Figures 5 and 6 show primary sensor (XQ2) temperature and relative humidity profiles, respectively, for all the flights conducted between 15-19 July 2018. The profiles are plotted using an artificial horizontal axis offset for clarity. These figures serve the purpose of a quick glance over the entire dataset and to locate interesting flights for further study. It should be noted that all the presented data are raw data as collected by the sensors without any correction for sensor response time or biascorrection. In Figure 5, flights conducted on 15, 16, and 18 July to investigate 'Convection initiation (CI)' show a well-mixed atmosphere profile for most flights with a steady lapse rate of temperature. Data from M600P1 on 18 July at the Golf location (see Table 2 and 3) show the presence of an inversion in the early morning flights. Also, notice the last ten profiles for M600P1 with varying speed produces an ascent-descent difference of various amounts due to change in effective sensor response time. Data collected at Leach airport to investigate 'Boundary layer transition (BLT)' on 17 July show a strong presence of an inversion in all flights.Data from 19 July collected to investigate 'Cold air drainage flow (CDF)' show progression of the ABL from inversion before sunrise in the early flights to well-mixed condition for the last few flights of the day.

In Figure 6, flights conducted on 17 July by M600P1 show primary humidity sensor failure. However, data files include secondary sensor humidity measurements that should be used for analysis instead. Since the humidity sensors have a higher sensor response time in the temperature we conducted most of our flights, it may show hysteresis higher than the temperature.We also found that the humidity sensor would collect micro dust particles as it was being flown, which could affect the accuracy of the sensors further. Another interesting feature of the humidity data presented here shows that readings are much smoother when collecting data in an inversion compared to data in a well-mixed atmosphere. Additionally, the difference between ascent and descent is much higher near ground level for most flights; this is the result of a rapid change of humidity near ground and sensor response time of humidity sensors." pg 10, line 185: Can the author contributions be more detailed? See other papers in the special issue for examples to consider.

Authors response:

We have updated the author contributions with more details as requested. It now reads:

"AH, and CD planned the contribution of the University of Nebraska-Lincoln contributions to LAPSE-RATE. AI designed the sensor housing and support structures. All authors contributed to data collection and analysis. AI, AS, and CD were part of the multirotor flight team. AI and AS contributed to data processing and presentation. AI constructed the manuscript. All authors contributed to manuscript edits. AH, and CD acquired the funding for the paper."

**Response to Anonymous Referee #1 comments**

**General Comments from Anonymous Referee #1:**

This data paper highlights contributions from researchers at the University of Nebraska-Lincoln to the 2018 LAPSE-RATE Campaign. The paper is well organized and provides sufficient explanation of the hardware used in data collection. The sensor deployment used to collect the data is relatively unique among LAPSE-RATE participants.

```
Authors response:
```

Thank you for your detailed feedback and comments to improve the quality of the paper. We have made the suggested changes and corrections to the paper.

**Specific Comments:**

Line 17 - Define what fixed-site profiling means here. I believe the intent is that multirotors can fly to and remain at a fixed point for a period of time. Authors response:

We have used a different word to better express the intent. The line 17 (now 35) reads: " Multirotors extend the sampling capability by allowing rapid and repeatable profiling at any site while maintaining a fixed horizontal position."

Line 20 - Define CLOUD-MAP before first use.

Authors response:

We have added a definition for the term before its first use.

Line 50 - Why weren't the barometric pressure sensors integrated in the iMet-XQ2 and nimbus-pth sensors included in the technical description?

Authors response: We have added a note to indicate that pressure sensor data is also available. Our sensor housing design is focused on the temperature and humidity sensors and as such the technical discussion is focused on that. However, we included the pressure sensor data that we collected so that can be used by anyone interested (e.g., to find potential temperature from temperature data).

Line 82 - Elaborate on what periodic checks of the data means. I assume you mean a human is observing that data are being collected and the values appear to be reasonable. Were there instances where you observed abnormal data collection mid-flight using the wireless data stream and modified or aborted a flight, or switched

**out instrumentation after a flight?**

Authors response:

We clarified what we meant by periodic checks. We did not observe the data in real time but only after the UAS has landed. The text now reads: "The Odroid was connected with a ground computer using wireless 2.4 GHz XBee radios for the operation of DAQ, debugging, and periodic checks on the data when the UAS finished a flight. The data collected by the DAQ were retrieved to the ground computer for archiving at the end of each day using an ethernet connection."

Line 83 & 84 - Was the interface between the DJI M600P flight controller and the Odroid a turn-key solution or did you have to develop any custom software to decode the DJI telemetry stream? Were GPS data from the iMet-XQ2/1 discarded?

Authors response:

We have added more information on this in the text. We used DJIs developer API for ROS to enable streaming of telemetry data to the DAQ. We didn't write any custom software for DJI flight controllers. GPS data from iMets were discarded during the LAPSE-RATE data collections as we found it to be very unreliable. But the software was later fixed after the campaign to enable recording of GPS data as additional redundancy. We added the following text:

"The communication with the DJI flight controller was implemented using the ROS interface of DJI Onboard SDK (DJI,2021c) available to developers. This allowed the recording of all the telemetry data from the flight controller, along with high-quality positioning information. The GPS data from the iMet XQ2/ iMet XQ1 sensor were discarded as the positioning information from the flight controller was found to be of better quality."

Line 111 - Define MURC before first use.

Authors response:

We have added a definition for the term before first use.

**Technical Corrections:**

Line 83 - add "The" in front of "UAS's".

Authors response:

Thank you for pointing it out. We have fixed this error.

Lines 102 & 103 - Spell out approximate instead of approx. Replace ~ with the same term for consistency. Authors response:

Thank you for pointing it out. We have replaced the sign and short notation.

Use "g" instead of "gm" for abbreviated units of grams.

Authors response:

Thank you for pointing it out. We have fixed this error and other unit inconsistencies throughout the paper.

Check the journal spacing requirements when using units. Spacing is inconsistent throughout the manuscript and tables.

Authors response:

Thank you for pointing it out. We have fixed the unit spacing according to journal requirements throughout the paper.

Some minor editing for consistent uses of past tense is needed.

Authors response:

Thank you for pointing it out. We have edited the paper to make more consistent use of past tense.

**General Comments from Anonymous Referee #2:**

This data overview paper outlines and describes the rotary-wing UAS data collected by the University of Nebraska-Lincoln in the 2018 LAPSE-RATE campaign. The writing is clear and concise and the paper is decently well structured. There are a few aspects that I would like to see improved before publication, and I move to accept with major revisions to enhance the details of this paper.

I think that the discussion of the hardware (Section 2), though concise, was well handled and provides a thoughtful overview of the system utilized for data collection. However, more information about the logistics of data collection (Section 3) would be nice.

The biggest complaint that I have with this paper is that it is lacking in context and specifics. This paper feels detached from the special issue's context. It currently does not even reference the campaign's overview article in this special issue or the Bulletin of the American Meteorological Society that would help provide the missing backdrop for this data set. Please make a greater effort to tie in your work to the context of the larger effort.

In general, there are a few stylistic points that could be improved as well. Most figure captions are also lacking in detail that could help to better inform the reader about the purpose of including the figures. This issue should also be addressed in the main text by discussing the figures and their significance more. Moreover, please be sure to follow the ESSD journal conventions for including numbers and units (see here: https://www.earth-system-science-data.net/submission.html#math). Please see the points below for more specific instances of these recommendations.

**Authors response:**

Thank you for your detailed feedback and comments to improve the clarity of the texts, inclusion of contexts for a broader audience, and quality of the paper. We also appreciate the feedback to extend the paper to improve the relationship of the paper to the overall LAPSE-RATE campaign.

**Major Comments**

- Section 1, Introduction: in general, this paper is missing the context of being part of the larger LAPSE-RATE campaign, which should be improved by including references and discussion for at least the following:
  - a. de Boer et al. (2020a): https://doi.org/10.1175/BAMS-D-19-0050.1
  - b. de Boer et al. (2020b): https://doi.org/10.5194/essd-12-3357-2020

Authors response:

We have expanded the introduction, added more citations including the one mentioned, and added more text throughout the paper describing our data collection effort in the context of the larger LAPSE-RATE campaign. We have also added citations for other platforms collecting data simultaneously with us in some occasions (such as calibration or inversion flights). Additionally, -Added more context and citation in Section 3.1 describing our data collection sites in the context of LAPSE-RATE campaign. Updated figure 2 with an inset showing UNL's flight location in the context of all the LAPSE-RATE campaign team locations. -Added more details throughout Section 3.2 to highlight how the data collection of UNL was part of a collaborative effort between LAPSE-RATE teams.

- Sections 1 and 2.3: while the authors do a good job of discussing their sensor housing setup, it is also important to include references to other studies that have performed similar work to provide context to someone trying to use this data that may or may not be familiar with UAS sensor housing and limitations. I would therefore like to see the following:
  - a. Provide more discussion/details here from the the Islam et al. (2019) paper to better contextualize this specific aircraft.
  - b. Depending on the specific details you include from the Islam et al. (2019) study, it would also be beneficial to elaborate more on the Villa et al. (2016), and Prudden et al. (2016) studies.
  - More context could be added by including and possibly briefly discussing the Greene et al.
     (2019) study (https://doi.org/10.3390/s19061470), which is effectively a continuation of their
     2018 study you already cited and is more closely related to the applications of the sUAS discussed in this ESSD paper.

Authors response:

We have elaborated discussion of sensor housing setup and its difference with other relevant setups with more citations in both Section 1 and 2.3.

3. Section 3.1 (L 107): The reader has no context for what these sites are without the introduction of the LAPSE-RATE campaign as a whole, which is currently missing from Section 1. Please provide proper context (and citations) to how these sites fit with the larger campaign as well as some details about them.

Authors response:

Thank you for your suggestion. We have added more context throughout the paper to identify our work in the context of the larger LAPSE-RATE campaign as outlined in response to major comment 1.

4. Section 3.2: I think this section would benefit from further organization, more specifically by splitting each day into subsections and provide more details per day (please look at the other accepted/published papers in this ESSD special edition).

Authors response:

Thank you for your recommendation. We have split the section into subsections and added more details per day to better inform the reader about the strategy and timeline of events on each day on each sampling location. 5. L 142: You mention using a "zero-order-hold-method" here as your main data processing method. Please elaborate what this method is and how you applied it here in constructing your data files.

Authors response:

We have elaborated what zero-order-hold means and justification for using it in our case. The text now reads: "....to match the output rate of primary sensors. In the ZOH method, sample value is held constant for one sampling period, i.e., when temperature data is recorded from temperature sensors, the last known value of altitude from GPS data is recorded without any interpolation. Since the GPS data is recorded at a higher frequency from the flight controller, it is assumed to be close and within GPS's uncertainty of measurement. Invalid or missing...."

6. Section 3: In Line 124, you mention you could not go to altitudes above 120 m because the NOTAM was not active. This sentence is a bit misleading because there is no mention of how else you accessed the airspace and generally the NOTAM is just the proof the request was filed properly. This section would benefit from what permissions you relied on (COA or 107 exemption) and what your maximum allowed altitude was. With more people getting into sUAS work, it's important for people within our community to be transparent about how to legally do this work.

Authors response:

Thank you for pointing it out. We have added additional description on the flight permission from FAA and our safety practices. The text in section 3.2 now reads:

"All the flights were conducted under the command of one remote pilot in command (PIC) with 'Federal Aviation Admin-istration (FAA) part 107' license in accordance with FAA's rule. All the flights included in the dataset were conducted using preprogrammed missions in DJI Ground Station (GS) Pro app (DJI, 2021a) by the remote pilot in command (PIC), with very few exceptions of manual flights. Occasionally the remote PIC took control over segments of flight from the automatic mis-sion control of the app when deemed safer by the PIC, e.g., passing through a turbulent layer of atmosphere. Although visual observers (VO) were not required by FAA, two VO were present at each flight location for greater situational awareness and safety during each flight. VOs were monitoring the UAS's movement, took handwritten notes about flight events and weather, and scanned the surrounding area for manned and unmanned flights.All the flights were legally conducted under FAA Certificate of Authorization (COA) for altitudes up to 914.4 m AGL when notices to airmen (NOTAMs) were active in the blue area marked in the 'inset (A)' of Figure 2. For all our flights, however, we were limited to flying up to a 500 m maximum altitude due to the altitude limitation set in the firmware of the UAS. In the days when NOTAMs were not active for COA, all the

flights were conducted up to the legal flight limit of 121 m AGL as defined in the 'part 107' regulations."

7. Section 5 Special topics of interest (L 152): I very much enjoy the presentation of how this data set can be used to examine broader questions of platform and sensor performance. However there is little to no context as to why these topics are important for people outside our community who might be interested in using this as sounding data. I suggest you provide few sentences of background as to why each of these are important before saying which flights may be utilized to examine this phenomena. For example, what are some of the challenges associated with not optimizing ascent/descent speeds? Wind direction versus sampling?

**Authors response:**

Thank you for your comment. We have added additional explanation and backgrounds on each section. The following texts were added for each subsection.

"5.1 Calibration....Correction of bias in sensor readings during post-processing requires calibration against a known reliable measurement. It also serves as additional validation for the sensor platforms and their collected data. It also facilitates the com-parison of data collected by different platforms by providing a "ground-truth" to compare against. ...."

"5.2 Effect of ascent/descent speed.....While it is desirable to move at a faster speed to optimize battery power usage to profile at greater altitudes, it may contribute to the effective sensor response time. Characterizing the sensor response at the different ascent and descent speeds would allow for the corresponding correction in the post-processing of the data..."

"5.3 Detection of Inversion......The flightswere coordinated with radiosonde launches from National Severe Storms Laboratory (NSSL) to compare the UAS profiles against the radiosonde profiles. University of Nebraska-Lincoln (UNL) Mobile Mesonet was also collecting data at the ground for surface-level observations. Dataset for radiosonde observations by NSSL (Bell et al., 2021), and surface observations byUNL Mobile Mesonet (de Boer et al., 2020c) is uploaded to Zenodo for intercomparison. The ability to detect the inversion at the correct altitude by the UAS sensor proves that UAS is collecting the observations at the sensor level rather than from the upwash or downwash of the UAS. Additionally, detection of inversion provides confidence in the quality of the data from the sensor housing in both ascent and descent. Different ascent descent speeds are used to identify the maximum speed that can be used while still acquiring quality data. Characterization of the sensor in the inversion layer provides a means for correction of observation level in case an offset is detected in the inversion layer when compared to a radiosonde. These data could also be used for comparison to the

11

theoretical work for ascent and descent rate of sensing platforms (Houston and Keeler, 2020)."

**Effect of body-relative wind direction / Horizontal transect:**

"5.4 Effect of body-relative wind direction and Horizontal transect.....These data can also be compared with radiosonde profile (Bell et al., 2021) and surface observations (de Boer et al., 2020c) similar to Section 5.3. The horizontal flights at different speeds against various orientations of wind provide additional characterizations for the quality of sensor data at various atmospheric wind conditions. Different horizontal flight speed simulates different incident wind speed at the sensor housing inlet and their effect on the observations. At the same time, the orientation of sensor housing simulates incident wind at different orientations and their effects on the sensor observations. The orientation characterization is particularly important as waste heat from UAS can be carried into the sensor housing in an unfavorable wind orientation. Any bias that may appear in these tests would need to be considered in the profiling flight plan to optimize the orientation of the sensor housing inlet relative to the wind to collect quality data and make appropriate corrections in the post-processing. Our analysis of these data can also be found in our previous work (Islam et al., 2019). Although traditionally multirotor UAS is used for vertical profiling; our data shows reliable data collection is also possible for horizontal profile/transect using our sensor housing. "

8. Figures 4 and 5: I'm not sure I understand the utility of presenting the data in this manner. Especially in the humidity sensors you seem to be having a fair bit of hysteresis. Please provide discussion as to whether this is an accurate depiction of the environmental variability or if it more closely linked to sensor hysteresis.

Authors response:

Thank you for your comment. We have clarified in the caption and description in figure 4 and 5 (now 5 and 6) that the data we presented are raw data and not corrected for sensor bias. Since the humidity sensor has much slower response time hysteresis seen in raw data is higher as well. Additionally, we have added some discussion on results to notice on both figures that could be used to identify specific profiles to investigate further based on the science objective.

**Minor Comments:**

Physical unit convention requirements:

- 1. L 44: there should be a space between the number and units; "mm" should not be italicized
- 2. L 47: change instances of "m/s" to use a "-1" exponent instead

- 3. L 55: insert a space between "1 Hz"
- 4. L 70: unit conventions on 5m/s
- 5. L 103: fix the unit conventions for "grams"
- 6. L 113-114: fix the unit conventions for m, m/s, and seconds.
- 7. L 119: fix the unit conventions for m and m/s
- 8. L 124: fix unit conventions for m and m/s
- 9. L 143: "1 second" should be "1 s" to be consistent with unit conventions
- 10. L 161-162: unit conventions for m and m/s
- 11. L 165: unit conventions for m/s

**Authors response:**

Thank you for pointing it out. We have fixed the unit spacing and presentation according to journal requirements throughout the paper.

**Grammar issues:**

- 1. L 8: replace "temperature/humidity sensors" with "temperature and humidity sensors"
- 2. L 8: I believe there is a subject/verb disagreement in "...attempt to separate them..."
- 3. L 20: please define the acronym "CLOUD-MAP"
- 4. L 54: remove the comma after "GPS" at the start of the line
- 5. L 56: please define the acronym "DAQ" as this is the first instance of use
- 6. L 61: insert "the" so it reads: "In the data files, the first two sensors..."
- 7. L 79: 'Fail' should be 'failure'.
- 8. L 81: insert a comma to read as: "...DAQ, debugging, and periodic..."
- 9. L 84: 'Interface' should be 'interfaces'.
- 10. L 100: add an "s" onto the end of "sensor" to read as: "It also allows comparison of the sensors mounted on..."
- 11. L 101: Add a comma after "when the primary sensors fail,"
- 12. L 101: Remove the comma in the date at the end of the line to read as "17 July 2018".
- 13. L 105, Section 3 header: remove the comma after "locations"
- 14. L107: add an "and" and "the" to read as: "...Leach, India, and Charlie in the LAPSE-RATE flight campaign".
- 15. L 119: change "in Golf and Gamma location" to "at the Golf and Gamma locations"
- 16. L 127: add an "s" at the end of "condition"
- 17. L 141: add "the" in front of "UAS flight controller"
- 18. L 160: date format earlier in the paper was DD month YYYY; please change to be consistent here
- 19. L 161: add a local conversion to MDT from UTC
- 20. L 174: change wording at the beginning to be: "Figures 4 and 5 show primary sensor..."
- 21. Table 3: No. of Flight should be plural

**Authors response:**

Thank you for pointing out the grammatical errors. We have fixed the errors and made additional grammar checks on the new revision of the paper.

**Formatting issues:**

- 1. L 50, Section 2.2 header: remove the colon at the end
- 2. Section 2.5 in general: please make the boldface headers into subsubsections. For example, L 91 should change to: "2.5.1 UAS platform M600P1", etc.
- 3. Section 5 in general: please specify subsections for the bold headings and remove the colons after the headings. For example, L 154 should read as: "5.1: Calibration".
- 4. Table 3: Please make more noticeable if multiple aircraft are at the same location for a day (e.g., add an "&" in between them)

**Authors response:**

Thank you for pointing out the formatting issues. We have made the changes as requested to increase readability of the information presented in the paper.

**Other minor comments:**

1. L 14-15: more references providing examples of how multirotor UASs are gaining popularity would be useful

```
Authors response:
We added more citations with more descriptive multirotor examples.
Text now reads:
"Multirotor UASs are finding more routine uses for sampling and
profiling the ABL, such as atmospheric profiling (Bonin et al., 2013;
Elston et al., 2015; Greatwood et al., 2017; Jacob et al., 2018;
Islam et al., 2019; Barbieri et al., 2019; Segales et al., 2020),
estimation of the spatial structure of temperature (Hemingway et al.,
2020), wind measurement (Prudden et al.,2016; Palomaki et al., 2017),
and prediction of Lagrangian coherent structure (Nolan et al.,
2018)."
```

 L 16-16: What are some examples of applications that would benefit from sounding data with increased spatiotemporal resolution? Additional references and specifics here would be nice. Authors response:

We added more citations with examples of applications that would benefit from increased resolution. The text now reads: "The need for increased spatial resolution for atmospheric sampling is reflected in publications, such as improving Numerical Weather Prediction (NWP) models (Leuenberger et al., 2020), improvement of mesoscale atmospheric forecast (Dabberdt et al., 2005), and identification of hazardous weather for Beyond Visual Line of Sight(BVLOS) flights using UAS Traffic Management (UTM) systems (Mitchell et al., 2020)."

3. L 25: what do you mean by "validity of the measurement"? Please elaborate. Authors response: We changed the wording from 'validity' to 'accuracy' to improve the clarity of meaning as we intended in the text.

4. L 39: to conclude the introduction, please include a general outline for how the rest of the paper is organized.

Authors response: We added a general outline of how the rest of the paper is organized.

- 5. L 90: please connect these sensor descriptions back to the diagram of the UAS in Figure 1 for reference. Authors response: We added more sentences referencing the sensor positions in Figure 1.
- 6. L 113-114: "ascended to the height of the MURC tower." How tall is that?

Authors response: We have clarified the height of the MURC tower. We also added additional details for other platforms collecting data simultaneously. We have also added an image of the UAS platforms sampling next to the MURC tower to add visual context for the reader about spatial distribution of the sensing platforms. Text now reads:

"MURC tower instrumentations were set to 15.2 m AGL. University of Nebraska-Lincoln (UNL) Mobile Mesonet was also collecting data about 2 m AGL for surface-level observations. Additionally, periodic radiosonde launches were conducted by National Severe StormsLaboratory (NSSL). Figure 3 shows an overview of the spatial distribution of the MURC tower, UAS platforms, and UNLMobile Mesonet. Details about MURC tower's instrumentation, deployment strategies, and data processing can be obtained from (de Boer et al., 2020c)..

. . . . . . .

The data are available for the MURC tower (de Boer et al., 2020c), UNL Mobile Mesonet (de Boer et al., 2020c), radiosonde (Bell et al., 2021), and all other participating teams on 14 - 15 July 2018 in the Zenodo community for LAPSE-RATE at (LAPSE-RATE Data Repository, 2021)."

7. L 120-121: Please split up the weather descriptions here and for the other days into multiple complete sentences.

```
Authors response:
We split up the weather descriptions into multiple complete sentences
to improve readability.
```

 L 150: please describe the file conventions in detail here even though it is also included in the README file to be consistent with the other published/accepted papers in this ESSD special issue. Authors response:

Thank you for your recommendation. We added the file convention in more detail here with added citation for further reading. The text now reads:

"Files were formatted in NetCDF format, with common variables names and meta-data added, to be consistent with all the entities collecting data for the LAPSE-RATE field campaign. A detailed explanation of the naming conventions and meta-data that were requested can be obtained from (de Boer et al., 2020b). An example file name produced by UAS platforms M600P1, and M600P2 for the data collected starting at 23:16:33 UTC on 14 July 2018 would be UNL.MR6P1.a0.20180714.231633.nc, and UNL.MR6P2.a0.20180714.231633.nc respectively. Here,

-`UNL' is the identifier for the data collecting institution, UNL -`MR6P1', and `MR6P2' are the platform identifiers for M600P1, and M600P2 respectively -`a0' indicates raw data converted to NetCDF -`20180714' is UTC file date in yyyymmdd(year, month, day) format -`231633' is UTC file start time in hhmmss(hours, minutes, seconds) format -`nc' is the NetCDF file extension

All the files also contain metadata for each variable with an explanation of physical measurement units, time synchronization method, sensors used for the measurement. File naming conventions and explanations are also described in the read-me file of the Zenodo data repository."

- 9. L 171-173: this information more appropriately belongs in the caption for Figure 3 Authors response: We have added the information in the caption for Figure 3 (now Figure 4).
- 10. L 195, References section: please alphabetize your references Authors response: Thank you for pointing it out. We have alphabetized the references according to the ESSD guidelines.
- 11. Figure 1: additional close-up photos of the shields and under-body should be included to give the reader a better spatial understanding of the UAS sensor payload

Authors response: Thank you for the suggestion. We added three additional close-ups pictures as panels. We have improved the caption with more description as well to better inform the reader. The caption now reads:

"Images of (A) the UAS setup with the temperature and humidity sensor mounted in the aspirated and shielded sensor housing, and in a traditional configuration (B) Close up of the traditionally mounted sensor under the UAS without the sensor housing (inside the whitecircle), (C) Close up of the sensor housing with the sensor mounted, and (D) Close up of the sensor probe mounted inside our sensor housing. The outlet of the sensor housing is placed on top of the propeller, and the inlet is pointing outward. High-speed air in the sensor housing is drawn passively by exploiting the pressure deficit created by the propeller of the UAS."

12. Figures 3-5: It would be helpful to label your panels A, B, C, D, etc. and then further describe the nuances in the captions.

Authors response: We added more detail in the captions to describe the figures to aid readers with synthesis of information from the figures.

**University of Nebraska UAS profiling during LAPSE-RATE**

Ashraful Islam1, Ajay Shankar1, Adam Houston2, and Carrick Detweiler1

1NIMBUS Lab, Department of Computer Science & Engineering, University of Nebraska-Lincoln, Lincoln, NE 68588, USA. 2Department of Earth & Atmospheric Sciences, University of Nebraska-Lincoln, Lincoln, NE 68588, USA.

Correspondence: Ashraful Islam (mislam@huskers.unl.edu)

**Abstract.**

[revised manuscript text omitted]

- 40 from the UAS to sample just outside of the UAS turbulence in both ascent and descent. This is different from existing methods of placing the sensor under the arm without shielding but aspirated by the propeller (Hemingway et al., 2017), on the body of the UAS without shielding and aspiration (Lee et al., 2018), on a different part of the UAS with shielding and possible aspiration from propellers (Greene et al., 2018) or shielding the sensor inside UAS and active aspiration using a fan while pointing the inlet towards the wind (Greene et al., 2019). All of these existing configurations fail to produce reliable data
- 45 during descent, and these data are usually discarded (Lee et al., 2018). As multirotor flight time is very limited, needing to discard entire descent data prevents optimal use of resources. Additionally, in most cases, observations are affected by wind direction and require onboard sensing of wind and reorientation of UAS with the change of wind direction (Greene et al., 2019). The sensor housingdesign has evolved over multiple design iterations and has been field tested in multiple CLOUD-MAP field campaigns (Jacob et al., 2018; ?).
- 50 Two primary highlights of the our novel sensor housing are the ability to reliably obtain sensor reading-its ability to obtain temperature and humidity sensor readings reliably during both ascent and descent profiles, and its invariance to the aircraft orientation relative to the ambient wind. Two key design considerations to achieve in achieving these goals are: the placement of the sensor<del>and, and its consistent aspiration.</del> Placement of the sensor on the UAS body can adversely affect the measurements (Greene et al., 2018; Jacob et al., 2018). According to experimental resultspresented by As observed through
- 55 prior experimental results (Villa et al., 2016), the validity of the measurement increases farther away accuracy of a sensor's measurement increases the farther away it is placed from the propeller's downwash. More specificallyfrom (Prudden et al., 2016), sensors placed at least 2.5×, a sensor placed at a distance at least 2.5 times the propeller diameter away from the rotor experiences significantly less propeller interferenceaerodynamic interference (Prudden et al., 2016). Consistent and sufficient aspiration is also necessary for a consistent effective sensor response time (Houston and Keeler, 2018). Placing the sensor inside the
- 60 propeller region or near the body can result in inconsistent aspiration due to rotor turbulence (?Yoon et al., 2017)(Diaz and Yoon, 2018; Yoo